# Kurilosides A_1_, A_2_, C_1_, D, E and F—Triterpene Glycosides from the Far Eastern Sea Cucumber *Thyonidium (= Duasmodactyla) kurilensis* (Levin): Structures with Unusual Non-Holostane Aglycones and Cytotoxicities

**DOI:** 10.3390/md18110551

**Published:** 2020-11-06

**Authors:** Alexandra S. Silchenko, Anatoly I. Kalinovsky, Sergey A. Avilov, Pelageya V. Andrijaschenko, Roman S. Popov, Pavel S. Dmitrenok, Ekaterina A. Chingizova, Vladimir I. Kalinin

**Affiliations:** G.B. Elyakov Pacific Institute of Bioorganic Chemistry, Far Eastern Branch of the Russian Academy of Sciences, Pr. 100-letya Vladivostoka 159, 690022 Vladivostok, Russia; sialexandra@mail.ru (A.S.S.); kaaniv@piboc.dvo.ru (A.I.K.); avilov-1957@mail.ru (S.A.A.); pandryashchenko@mail.ru (P.V.A.); prs_90@mail.ru (R.S.P.); paveldmt@piboc.dvo.ru (P.S.D.); martyyas@mail.ru (E.A.C.)

**Keywords:** *Thyonidium kurilensis*, triterpene glycosides, kurilosides, sea cucumber, cytotoxic activity

## Abstract

Six new monosulfated triterpene tetra-, penta- and hexaosides, namely, the kurilosides A_1_ (**1**), A_2_ (**2**), C_1_ (**3**), D (**4**), E (**5**) and F (**6**), as well as the known earlier kuriloside A (**7**), having unusual non-holostane aglycones without lactone, have been isolated from the sea cucumber *Thyonidium (= Duasmodactyla) kurilensis* (Levin) (Cucumariidae, Dendrochirotida)*,* collected in the Sea of Okhotsk near Onekotan Island from a depth of 100 m. Structures of the glycosides were established by 2D NMR spectroscopy and HR-ESI mass spectrometry. Kurilosides of the groups A and E contain carbohydrate moieties with a rare architecture (a pentasaccharide branched by C(4) Xyl1), differing from each other in the second monosaccharide residue (quinovose or glucose, correspondingly); kurilosides of the group C are characterized by a unique tetrasaccharide branched by a C(4) Xyl1 sugar chain; and kurilosides of the groups D and F are hexaosides differing from each other in the presence of an *O*-methyl group in the fourth (terminal) sugar unit. All these glycosides contain a sulfate group at C-6 of the glucose residue attached to C-4 Xyl1 and the non-holostane aglycones have a 9(11) double bond and lack γ-lactone. The cytotoxic activities of compounds **1**–**7** against mouse neuroblastoma Neuro 2a, normal epithelial JB-6 cells and erythrocytes were studied. Kuriloside A_1_ (**1**) was the most active compound in the series, demonstrating strong cytotoxicity against the erythrocytes and JB-6 cells and a moderate effect against Neuro 2a cells.

## 1. Introduction

Triterpene glycosides isolated from different species of the sea cucumbers demonstrate promising biological activities [1,2] and a great structural diversity, including some recently found structural features unique for the glycosides [3,4,5] of terrestrial or marine origin. These glycosides usually form an extremely complicated mixture in the organism producer. So, its separation and isolation of dozens of pure individual compounds, especially minor ones, became possible only as a result of the development of chromatographic equipment and methodology. Thus, the glycoside compositions of some sea cucumber species were reinvestigated during last years, and novel minor glycosides were found, and the structures of some known substances were corrected [2,5,6,7].

The reinvestigation of the glycoside composition of the sea cucumber *Thyonidium* (= *Duasmodactuyla*) *kurilensis* (Levin) was undertaken for the same reason. Earlier studies of the glycosides of this species showed the complexity of its glycoside mixture. So, a part of the glycoside sum was subjected to acid hydrolysis followed by the separation of the obtained derivatives. As a result, the structures of two genins, kurilogenin [8] and nemogenin [9], were established. Later, two glycosides, kurilosides A (**7**) and C, were isolated [10]. The absolute configurations of the monosaccharide residues, composed of the carbohydrate chains of kurilosides A (**7**) and C, were assigned as *D* [10]. However, the remaining part of the glycoside fraction, containing more polar and minor glycosides, remained unexplored.

Herein, we report the isolation and structure elucidation of six glycosides, kurilosides A_1_ (**1**), A_2_ (**2**), C_1_ (**3**), D (**4**), E (**5**) and F (**6**), as well as of the known kuriloside, A (**7**), obtained in native form from the glycoside mixture. The animals were collected near Onekotan Island in the Sea of Okhotsk. The structures of the novel compounds **1**–**6** were established and the structure of **7** was corroborated through analyses of the ^1^H, ^13^C NMR, 1D TOCSY and 2D NMR (^1^H,^1^H-COSY, HMBC, HSQC and ROESY) spectra, as well as through HR-ESI mass spectra. All the original spectra are presented in Appendix A. The hemolytic activities against mouse erythrocytes and the cytotoxic activities against mouse neuroblastoma Neuro 2a and normal epithelial JB-6 cells were also studied.

## 2. Results and Discussion

### 2.1. Structural Elucidation of the Glycosides

The concentrated ethanolic extract of the sea cucumber *Thyonidium (= Duasmodactyla) kurilensis* was chromatographed on a Polychrom-1 column (powdered Teflon, Biolar, Latvia), repeated chromatography on Si gel columns, to give five fractions (I–V). Fraction I was subsequently subjected to HPLC on a reversed-phase semipreparative column to yield the compounds **1**–**6** (Figure 1) as well as kuriloside A (**7**), isolated earlier from this species.

The ^1^H and ^13^C NMR spectra corresponding to the carbohydrate chains of kurilosides A_1_ (**1**) and A_2_ (**2**) (Table 1) were identical to each other and to that of the known kuriloside A (**7**), isolated from this species earlier and repeatedly this time. The structure of the sugar chain of kuriloside A (**7**) was established earlier by the ^13^C NMR and chemical transformations (periodate oxidation, Smith degradation, Hakomori methylation followed by methanolysis, acetylation and GLC-MS analysis of the obtained products) [10]. The analysis of the 2D NMR spectra of the carbohydrate chain of the kurilosides of group A (**1**, **2**, **7**) was made for the first time (Table 1) and the structure established earlier was confirmed.

In the ^1^H and ^13^C NMR spectra of the carbohydrate part of **1**, **2** and **7**, five characteristic doublets at δ_H_ = 4.69–5.25 (*J* = 7.2–8.7 Hz), and corresponding to them signals of anomeric carbons at δ_C_ = 103.7–105.3, were indicative of a pentasaccharide chain and *β*-configurations of the glycosidic bonds. The ^1^H,^1^H-COSY, HSQC and 1D TOCSY spectra of **1**, **2** and **7** showed the signals of an isolated spin systems assigned to one xylose, one quinovose, two glucoses and one 3-*O*-methylglucose residue, which coincided with the monosaccharide composition of kuriloside A (**7**) established by chemical modifications [10]. The signal of C(6) Glc4 was observed at δ_C_ = 67.3, due to α-shifting effect of a sulfate group at this position.

The positions of the interglycosidic linkages were established by the ROESY and HMBC spectra of **1**, **2** and **7** (Table 1) where the cross-peaks between H(1) of the xylose and H(3) (C(3)) of an aglycone, H(1) of the second residue (quinovose) and H(2) (C(2)) of the xylose, H(1) of the third residue (glucose) and H(4) (C(4)) of the second residue (quinovose), H(1) of the fourth residue (glucose) and H-4 of the first residue (xylose), and H-1 of the fifth residue (3-*O*-methylglucose) and H-3 (C(3)) of the fourth residue (glucose) were observed, indicating the presence of branching by the C(4) Xyl1 pentasaccharide chain. Such an architecture is very rare for the sea cucumber glycosides, as is another finding—a carbohydrate chain of cladoloside J_1_ from *Cladolabes schmeltzii*, which, however, differed from kurilosides A (**7**), A_1_ (**1**) and A_2_ (**2**) in the monosaccharide composition [11].

The molecular formula of kuriloside A_1_ (**1**) was determined to be C_58_H_93_O_31_SNa from the [M_Na_ – Na]^−^ ion peak at an *m/z* of 1317.5449 (calc. 1317.5427) in the (–)HR-ESI-MS.

The analysis of the ^13^C and ^1^H NMR spectra of the aglycone part of **1** suggested the presence of an 22,23,24,25,26,27-hexa-*nor*-lanostane aglycone having a 9(11) double bond, which was deduced from the characteristic signals of the quaternary carbon C(9) at δ_C_ = 148.9 and tertiary carbon C(11) at δ_C_ = 114.0, with the corresponding proton signal at δ_H_ = 5.23 (brd, *J* = 5.6 Hz; H(11)) (Table 2). The signals at δ_C_ = 169.8 and 169.9 as well as the signals of the methyl groups at δ_C_ = 20.2 and 21.0 corresponded to the carbons of the two acetoxy groups, whose positions at C(16) and C(20) were established by the correlations between the protons of the *O*-acetate methyl groups (δ_H_ = 2.12 (s, COOC**H**_3_(16)) and H(16) (δ_H_ = 2.05 (s, COOC**H**_3_(20)) and H(20) in the ROESY spectrum of **1**. The HMBC correlations H(16) (δ_H_ = 5.64 (ddd, *J* = 5.2; 7.7; 13.4 Hz)/**C**OOCH_3_(16) and H(20) (δ_H_ = 5.46 (dd, *J* = 6.1; 10.6 Hz)/**C**OOCH_3_(20) corroborated these positions. Nemogenin—the aglycone with the same structure as in kuriloside A_1_ (**1**)—was obtained earlier as result of acid hydrolysis of the glycoside sum of *T. kurilensis* [9]. Nemogenin has a *β*-oriented *O*-acetic group at C(16), which was established by the comparison of the observed ^1^H NMR spectrum coupling constant *J*_16/17_ = 7.7 Hz with those calculated for the model 16*α*- and 16*β*-acetoxy-holostane derivatives as well as by the observed NOE between H(16α) and H(17α) [9]. The coupling constant (*J*_16/17_ = 7.7 Hz), observed in the ^1^H NMR spectrum of **1** (Table 2), coincided with that in nemogenin. The ROE correlation H(16)/H(32) corroborated the 16*β*-*O*Ac orientation. The (S)-configuration of the C(20) stereo-center in nemogenin was established by the analysis of inter-atomic distances in the models of the (20*R*)- and (20*S*)-isomers and the NOE-experiments. The correlations H(17)/H(21), H(20)/H(18) and H(18)/H(21) observed in the ROESY spectrum of **1** and the closeness of the coupling constant *J*_17/20_ = 10.6 Hz to that for nemogenin (*J*_17/20_ = 10.8 Hz) indicated the same (20*S*) configuration in kuriloside A_1_ (**1**).

The (–)ESI-MS/MS of **1** demonstrated the fragmentation of the [M_Na_–Na]^−^ ion at an *m/z* of 1317.5. The peaks of the fragment ions were observed at an *m/z* of 1257.5 [M_Na_–Na–CH_3_COOH]^−^, 1197.5 [M_Na_–Na–2CH_3_COOH]^−^, 1035.4 [M_Na_–Na–2CH_3_COOH–C_6_H_10_O_5_ (Glc)]^−^, 889.4 [M_Na_–Na–2CH_3_COOH–C_6_H_10_O_5_ (Glc)–C_6_H_10_O_4_ (Qui)]^−^ and 565.1 [M_Na_–Na–C_28_H_43_O_4_ (Agl)–C_6_H_10_O_5_ (Glc)–C_6_H_10_O_4_ (Qui)–H]^−^, corroborating the structure of kuriloside A_1_ (**1**).

All these data indicated that kuriloside A_1_ (**1**) is 3*β*-*O*-{*β*-d-glucopyranosyl-(1→4)-*β*-d-quinovopyranosyl-(1→2)-[3-*O*-methyl-*β*-d-glucopyranosyl-(1→3)-6-*O*-sodium sulfate*-β*-d-glucopyranosyl-(1→4)]-*β*-d-xylopyranosyl}-22,23,24,25,26,27-hexa-*nor*-16*β,(*20*S*)-diacetoxylanost-9(11)-ene.

The molecular formula of kuriloside A_2_ (**2**) was determined to be C_54_H_85_O_28_SNa from the [M_Na_–Na]^−^ ion peak at an *m/z* of 1213.4964 (calc. 1213.4954) in the (–)HR-ESI-MS.

Analysis of the ^13^C NMR spectrum of **2** indicated the presence of 22,23,24,25,26,27-hexa-*nor*-lanostane aglycone with the signals from C(1) to C(11), C(30), C(31) and C(32) close to those in the spectrum of **1** (Table 3). The signals of the olefinic carbons at δ_C_ = 144.5 (C(16)) and 152.1 (C(17)) with the corresponding olefinic proton H(16) at δ_H_ = 6.63 (brt, *J* = 2.6 Hz) indicated the presence of an additional double bond in the polycyclic nucleus of **2**. Its 16(17) position was deduced from the ^1^H,^1^H-COSY spectrum where the signals of protons H(15α)–H(15β)–H(16) formed an isolated spin system and was confirmed by the HMBC correlations: H(15α)/C(16, 17), H(15β)/C(16, 17), H(18)/C(17) and H(21)/C(17). The signal of the quaternary carbon at δ_C_ = 196.3 (C(20)) indicated the presence of a 20-oxo-group conjugated with a 16(17) double bond, which was confirmed by the correlations H(16)/C(20) and H(21)/C(20) observed in the HMBC spectrum of **2**. The structure of the aglycone of kuriloside A_2_ (**2**) was identical to that of the kurilogenin—an artificial genin—obtained from the glycoside sum of *T. kurilensis* as a result of acid hydrolysis. It was found first as a part of the native glycoside **2**.

The (–)ESI-MS/MS of **2** demonstrated the fragmentation of the [M_Na_–Na]^−^ ion at an *m/z* of 1213.5. The peaks of fragment ions were observed at an *m/z* of 1037.4 [M_Na_–Na–C_7_H_12_O_5_ (MeGlc)]^−^, 905.4 [M_Na_–Na–C_6_H_10_O_5_ (Glc)–C_6_H_10_O_4_ (Qui)]^−^, 565.1 [M_Na_–Na–C_24_H_35_O (Agl)–C_6_H_10_O_5_ (Glc)–C_6_H_10_O_4_ (Qui)–H]^−^, corroborating the identity of carbohydrate chains of **1** and **2**.

All these data indicated that kuriloside A_2_ (**2**) is 3*β*-*O*-{*β*-d-glucopyranosyl-(1→4)-*β*-d-quinovopyranosyl-(1→2)-[3-*O*-methyl-*β*-d-glucopyranosyl-(1→3)-6-*O*-sodium sulfate*-β*-d-glucopyranosyl-(1→4)]-*β*-d-xylopyranosyl}-22,23,24,25,26,27-hexa-*nor*-20-oxo-lanosta-9(11),16-diene.

The molecular formula of kuriloside C_1_ (**3**) was determined to be C_52_H_83_O_26_SNa from the [M_Na_–Na]^−^ ion peak at an *m/z* of 1155.4923 (calc. 1155.4899) in the (–)HR-ESI-MS. The ^1^H and ^13^C NMR spectra corresponding to the carbohydrate chain of kuriloside C_1_ (**3**) (Table 4) demonstrated four signals of anomeric doublets at δ_H_ = 4.66–5.13 (d, *J* = 6.9–8.2 Hz) and anomeric carbons at δ_C_ = 102.3–104.9 deduced by the HSQC spectrum. These data indicated the presence of a tetrasaccharide chain in **3**. Actually, its ^13^C NMR spectrum was similar with that of the known kuriloside C, isolated earlier from *T. kurilensis* [10], and different from the spectra of the carbohydrate part of kurilosides A (**7**), A_1_ (**1**) and A_2_ (**2**) by the absence of the signals corresponding to a glucose residue attached to C(4)Qui2 in their chain. The signal of C(4)Qui2 was shielded (δ_C_ 76.2) and the signals of C(3)Qui2 and C(5)Qui2 (δ_C_ = 76.7 and 72.9, correspondingly) were deshielded in the spectrum of **3**, when compared with these signals in the spectra of the kurilosides of group A (Table 1) due to the lacking of the glycosylation effects. Thorough analysis of the ^1^H,^1^H-COSY, the HSQC and 1D TOCSY spectra of **3** corroborated the presence of xylose, quinovose, glucose and 3-*O*-methylglucose residues. The positions of the interglycosidic linkages were elucidated based on the ROESY and HMBC correlations (Table 4). Hence, kurilosides C [10] and C_1_ (**3**) have a tetrasaccharide chain branched by C(4)Xyl1 and the part of the chain attached to C(2)Xyl1 consists of one monosaccharide only (quinovose), while the part attached to C(4)Xyl1 is composed of glucose, sulfated by C(6), and 3-*O*-methylglucose residues. This architecture of a carbohydrate chain is unique for the sea cucumber glycosides.

The NMR spectra of the aglycone part of kuriloside C_1_ (**3**) were coincident with those of kuriloside A_1_ (**1**), indicating the identity of their aglycones possessing two acetoxy-groups (Table 2).

The (–)ESI-MS/MS of **3** showed the fragmentation of the [M_Na_–Na]^−^ ion at an *m/z* of 1155.5. The peaks of the ions fragments were observed at an *m/z* of 1095.5 [M_Na_–Na–CH_3_COOH]^−^, 1035.5 [M_Na_–Na–2CH_3_COOH]^−^, 889.4 [M_Na_–Na–2CH_3_COOH–C_6_H_10_O_4_ (Qui)]^−^, 565.1 [M_Na_–Na–C_28_H_43_O_4_ (Agl)–C_6_H_10_O_4_ (Qui)–H]^−^, and confirmed the structure of **3**.

All these data indicated that kuriloside C_1_ (**3**) is 3*β*-*O*-{*β*-d-quinovopyranosyl-(1→2)-[3-*O*-methyl-*β*-d-glucopyranosyl-(1→3)-6-*O*-sodium sulfate*-β*-d-glucopyranosyl-(1→4)]-*β*-d-xylopyranosyl}-22,23,24,25,26,27-hexa-*nor*-16*β,(*20*S*)-diacetoxylanost-9(11)-ene.

The molecular formula of kuriloside D (**4**) was determined to be C_66_H_105_O_35_SNa from the [M_Na_–Na]^−^ ion peak at an *m/z* of 1489.6174 (calc. 1489.6163) in the (–)HR-ESI-MS.

The ^1^H and ^13^C NMR spectra corresponding to the carbohydrate part of kuriloside D (**4**) (Table 5) demonstrated six signals of anomeric doublets at δ_H_ = 4.70–5.28 (d, *J* = 7.5–8.2 Hz) as well as six signals of anomeric carbons at δ_C_ = 103.7–105.7 deduced from the HSQC spectrum, which indicated the presence of a hexasaccharide chain in **4**. The presence of xylose, quinovose, three glucose and 3-*O*-methylglucose residues was deduced from the analysis of the ^1^H,^1^H-COSY, HSQC and 1D TOCSY spectra of **4**. The positions of the interglycosidic linkages were elucidated based on the ROESY and HMBC correlations (Table 5). The comparison of the ^13^C NMR spectra of the carbohydrate chains of kurilosides A_1_ (**1**) and D (**4**) showed the coincidence of the signals assigned to the xylose, quinovose and glucose attached to C(4)Xyl1, sulfated by C(6), and the 3-*O*-methylglucose residues. The differences were observed between the signals assigned to the glucose, bonded to C(4)Qui2: the signal of C(3)Glc3 in the spectrum of **4** was deshielded (δ_C_ = 88.1) and the signals of C(2)Glc3 and C(4)Glc3 were shielded (δ_C_ = 73.6 and δ_C_ = 69.7, correspondingly) when compared with the spectrum of **1** (δ_C_ = 78.2 (C(3)Glc3), 74.7 (C(2)Glc3) and 71.4 (C(4)Glc3)). These shifting effects were observed due to the glycosylation of this glucose residue by the C(3) position with an additional glucose residue. Its signals were observed in the ^13^C NMR spectrum of **4** and its anomeric proton correlated with H(3)Glc3 in the ROESY spectrum of **4** and with C(3)Glc3 in the HMBC spectrum, corroborating the position of its glycosidic bond (Table 5). Therefore, one of the terminal monosaccharide residues in kuriloside D (**4**) has no *O*-methyl group at C(3) in contrary with the majority of known glycosides from the sea cucumbers. Thus, kuriloside D (**4**) contains a sulfated hexasaccharide chain, a new finding for the glycosides of sea cucumbers. Sulfated hexaosides were earlier isolated only from one holothurian species—*Cladolabes schmeltzii* [11]—but had another monosaccharide composition and sulfate group position.

The analysis of the ^1^H and ^13^C NMR spectra of the aglycone part of **4** suggested the presence of a lanostane aglycone containing a non-shortened side chain (30 carbons) with a 9(11) double bond, which was deduced from the characteristic signals of the quaternary carbon C(9) at δ_C_ = 149.0 and tertiary carbon C(11) at δ_C_ = 115.0, with the corresponding proton signal at δ_H_ = 5.35 (brd, *J* = 6.2 Hz; H(11)) (Table 6). A lactone ring was absent and the signal of the methyl group C(18) was observed at δ_C_ = 16.9. Two strongly deshielded signals at δ_C_ = 216.6 and 216.5 corresponded to carbonyl groups, whose positions were deduced as C(16) and C(22), correspondingly, based on the correlations H(15)/C(16), H(17)/C(16), H(21)/C(22) and H(23)/C(22) in the HMBC spectrum of **4**. The protons of side chain H(23)-H(24)-H(26)-H(27) formed an isolated spin system and the protons H(15α) and H(15β) correlated only to each other in the ^1^H,^1^H-COSY spectrum of **4**, which confirmed the presence of oxo-groups at C(22) and C(16). The signals of the olefinic carbons at δ_C_ = 145.5 (C(25)) and 110.0 (C(26)) indicated the presence of a terminal double bond. Therefore, a new triterpene non-holostane aglycone of kuriloside D (**4**) has a normal side chain, two double bonds and two oxo-groups.

The (–)ESI-MS/MS of **4** demonstrated the fragmentation of the [M_Na_–Na]^−^ ion at an *m/z* of 1489.6. The peaks of the fragment ions were observed at an *m/z* of 1349.5 [M_Na_–Na–C_8_H_13_O_2_+H]^−^, corresponding to the loss of the side chain from C(20) to C(27), 1187.5 [M_Na_–Na–C_8_H_13_O_2_–C_6_H_10_O_5_ (Glc)]^−^, 1025.4 [M_Na_–Na–C_8_H_13_O_2_–2C_6_H_10_O_5_ (Glc)]^−^, 879.4 [M_Na_–Na–C_8_H_13_O_2_–2C_6_H_10_O_5_ (Glc)–C_6_H_10_O_4_ (Qui)]^−^ and 565.1 [M_Na_–Na–C_30_H_45_O_3_ (Agl)–2C_6_H_10_O_5_ (Glc)–C_6_H_10_O_4_ (Qui)–H]^−^, which confirmed the aglycone structure and the sugar units sequence in the carbohydrate chain of **4**.

All these data indicated that kuriloside D (**4**) is 3*β*-*O*-{*β*-d-glucopyranosyl-(1→3)-*β*-d-glucopyranosyl-(1→4)-*β*-d-quinovopyranosyl-(1→2)-[3-*O*-methyl-*β*-d-glucopyranosyl-(1→3)-6-*O*-sodium sulfate*-β*-d-glucopyranosyl-(1→4)]-*β*-d-xylopyranosyl}-16*,*22-dioxo-lanosta-9(11),25-diene.

The molecular formula of kuriloside E (**5**) was determined to be C_54_H_87_O_29_SNa from the [M_Na_–Na]^−^ ion peak at an *m/z* of 1231.5082 (calc. 1231.5059) in the (–)HR-ESI-MS.

In the ^1^H and ^13^C NMR spectra of the carbohydrate part of kuriloside E (**5**) (Table 7), five signals of anomeric doublets at δ_H_ = 4.71–5.27 (d, *J* = 7.0–7.8 Hz) and corresponding to them signals of the anomeric carbons at δ_C_ = 104.0–105.4 deduced from the HSQC spectrum were observed. This indicated the presence of a pentasaccharide chain in **5**. The comparison of the ^13^C NMR spectra of the sugar parts of kuriloside A_1_ (**1**) and E (**5**) revealed the differences among the signals of the second monosaccharide residue, attached to C(2)Xyl1. The analysis of the ^1^H,^1^H-COSY, HSQC and 1D TOCSY spectra of **5** showed this residue is a glucose. The signals of the rest of the monosaccharide units were close in the ^13^C NMR spectra of **1** and **5**. The only sulfate group is attached to C(6)Glc4 (δ_C_ 67.5), as in all other glycosides from *T. kurilensis*. The positions of the interglycosidic linkages elucidated by the ROESY and HMBC correlations (Table 7) were the same as in the kurilosides of group A. Thus, kuriloside E (**5**) is a branched monosulfated pentaoside with three glucose residues in the oligosaccharide chain—one of them occupying the second position—instead of the quinovose residue in the carbohydrate chains of compounds **1**–**4** and **7** and the majority of the other glycosides from sea cucumbers.

The ^1^H and ^13^C NMR spectra of the aglycone part of kuriloside E (**5**) demonstrated the presence of a hexa-*nor*-lanostane aglycone, having a 9(11) double bond and lacking a γ-lactone, as with the other kurilosides A_1_–D (**1**–**6**) (Table 8). The oxo-group (signal at δ_C_ 208.8) was positioned as C(20) based on the correlations H(17)/C(20) and H(21)/C(20) in the HMBC spectrum of **5**. The comparison of the ^13^C NMR spectra of the aglycone parts of kurilosides A_2_ (**2**) and E (**5**) showed the similarity of the signals from C(1) to C(11) as well as the signals of the methyl groups C(30), C(31) and C(32) and the differences of the signals of the carbons assigned to ring D. This was explained by the absence of the second double bond in the aglycone of **5** in comparison with **2**. So, the aglycone of kuriloside E (**5**) was identical to that of isokoreoside A isolated first from *Cucumaria conicospermium* [12] and then found in *C. frondosa* [13].

The (–)ESI-MS/MS of **5** demonstrated the fragmentation of the [M_Na_–Na]^−^ ion at an *m/z* of 1231.5. The peaks of the ion fragments were observed at an *m/z* of 1069.5 [M_Na_–Na–C_6_H_10_O_5_ (Glc)]^−^, 1055.4 [M_Na_–Na–C_7_H_12_O_5_ (MeGlc)]^−^, 907.4 [M_Na_–Na–2C_6_H_10_O_5_ (Glc)]^−^ and 565.1 [M_Na_–Na–C_24_H_37_O (Agl)–2C_6_H_10_O_5_ (Glc)–H]^−^, which confirmed the presence of glucose as the second sugar unit in the carbohydrate chain of kuriloside E (**5**).

All these data indicated that kuriloside E (**5**) is 3*β*-*O*-{*β*-d-glucopyranosyl-(1→4)-*β*-d-glucopyranosyl-(1→2)-[3-*O*-methyl-*β*-d-glucopyranosyl-(1→3)-6-*O*-sodium sulfate*-β*-d-glucopyranosyl-(1→4)]-*β*-d-xylopyranosyl}-22,23,24,25,26,27-hexa-*nor*-20-oxo-lanost-9(11)-ene.

The molecular formula of kuriloside F (**6**) was determined to be C_61_H_99_O_34_SNa from the [M_Na–_Na]^−^ ion peak at an *m/z* of 1407.5778 (calc. 1407.5744) in the (–)HR-ESI-MS.

In the ^1^H and ^13^C NMR spectra corresponding to the carbohydrate part of kuriloside F (**6**) (Table 9), six signals of anomeric doublets at δ_H_ = 4.70–5.26 (d, *J* = 7.2–8.6 Hz) along with the signals of the corresponding anomeric carbons at δ_C_ = 103.7–105.5 were observed. This indicated the presence of a hexasaccharide chain in **6**. The comparison of the ^13^C NMR spectra of the sugar moieties of kurilosides D (**4**) and F (**6**) showed the coincidence of the signals of the five monosaccharide units, except for the signals of the terminal (fourth) residue. The analysis of the ROESY, ^1^H,^1^H-COSY, HSQC and 1D TOCSY spectra of **6** revealed the residue, attached to C(3)Glc3, to be 3-*O*-methylglucose, instead of a glucose in this position of the carbohydrate chain of **4**. The presence of two signals of the *O*-methyl groups at δ_C_ = 60.5 and 60.6 and at δ_H_ = 3.85 (s) and 3.86 (s) in the ^13^C and ^1^H NMR spectra of **6** as well as the shifting of the signal of C(3)MeGlc4 to 87.8 due to the attachment of *O*Me-group confirmed the presence of two residues of 3-*O*-methylglucose as the terminal units in the chain of kuriloside F (**6**). The positions of the interglycosidic linkages were elucidated based on the ROESY and HMBC correlations (Table 9).

The carbohydrate chain of kuriloside F (**6**) is a new for the sea cucumber glycosides. This is the fifth representative of the sulfated hexaosides along with kuriloside D (**4**) found in the sea cucumbers.

The analysis of the ^13^C and ^1^H NMR spectra of the aglycone part of **6** indicated the presence of 22,23,24,25,26,27-hexa-*nor*-lanostane aglycone, with a 9(11) double bond (Table 10). The deshielding of C(16) to δ_C_ = 71.1 and H(16) to δ_H_ = 5.40 (brt, *J* = 7.8 Hz) indicated the presence of a hydroxyl group at C(16), which was confirmed by the correlations H(15)/C(16) and H(17)/C(16) in the HMBC spectrum of **6**. The comparison of the NMR spectra of the aglycone parts of kuriloside F (**6**) and known kuriloside A (**7**) [10] showed their difference in the signals C(15), C(16) and C(17) due to the presence of different substituents (hydroxy or acetoxy group) at C(16). This was also corroborated by the (–)HR-ESI-MS spectra of **6** and **7**, differing by 42 *amu*, corresponding to a C_2_H_2_O fragment.

The ROE correlations H(16)/H(15β) and H(16)/H(18) indicated a 16*α*-OH orientation in the aglycone of kuriloside F (**6**). The comparison of the coupling patterns of the protons of ring D in kuriloside F (**6**) and the known earlier kuriloside A (**7**), having an α-oriented *O*-acetic group at C(16) [10], showed their closeness. Hence, the aglycone of **6** having an *α*-hydroxylated C(16) can be considered as the biosynthetic precursor of the aglycone of **7** characterized by the O-acetic group at this position. Moreover, the glycosides with the 16-hydroxy groups have never been isolated earlier from the sea cucumbers. It probably may be related to the unusual α-OH orientation at C(16) in the glycoside from *T. kurilensis* while the other known glycosides are characterized by the *β*-oriented 16-acetoxy group. Apparently, their biosynthetic 16*β*-hydroxylated precursors are quickly transformed in order to “protect” the aglycone against 18(16) lactonization (it is known that the simultaneous presence of hydroxyls at C-16 and C-20 in 18-carboxylated precursor preferably leads to formation of an 18(16) lactone [1]) for the holostane-type aglycones (having 18(20)-lactone) to be formed.

So, all these data indicated that the aglycone of kuriloside F (**6**) is 22,23,24,25,26,27-hexa-*nor*-16α-hydroxy-20-oxo-lanost-9(11)-ene, first discovered in the glycosides from the sea cucumbers, and can be considered as a “hot metabolite”, which is usually quickly metabolized into other derivatives and is the biosynthetic precursor of the 16-*O*-acetylated glycosides, particularly, kuriloside A (**7**).

The (–)ESI-MS/MS of **6** demonstrated the fragmentation of the [M_Na_–Na]^−^ ion at an *m/z* of 1407.5. The peaks of the ion fragments were observed at an *m/z* of 1231.5 [M_Na_–Na–C_7_H_12_O_5_ (MeGlc)]^−^, 1069.4 [M_Na_–Na–C_7_H_12_O_5_ (MeGlc)–C_6_H_10_O_5_ (Glc)]^−^, 923.4 [M_Na_–Na–C_7_H_12_O_5_ (MeGlc)–C_6_H_10_O_5_ (Glc)–C_6_H_10_O_4_ (Qui)]^−^ and 565.1 [M_Na_–Na–C_24_H_37_O_2_ (Agl)–C_7_H_12_O_5_ (MeGlc)–C_6_H_10_O_5_ (Glc)–C_6_H_10_O_4_ (Qui)–H]^−^, corroborating the structure elucidated by the NMR.

All these data indicated that kuriloside F (**6**) is 3*β*-*O*-{3-*O*-methyl-*β*-d-glucopyranosyl-(1→3)-*β*-d-glucopyranosyl-(1→4)-*β*-d-quinovopyranosyl-(1→2)-[3-*O*-methyl-*β*-d-glucopyranosyl-(1→3)-6-*O*-sodium sulfate*-β*-d-glucopyranosyl-(1→4)]-*β*-d-xylopyranosyl}-22,23,24,25,26,27-hexa-*nor*-16α-hydroxy-20-oxo-lanost-9(11)-ene.

Kuriloside A (**7**) was also isolated by us from the glycoside sum of *T. kurilensis* and identified with a known earlier compound [10] by the comparison of their ^1^H and ^13^C spectra. Moreover, extensive analysis of the 2D NMR spectra of **7** was made for the first time (Table 1 and Table 11). The positions of the interglycosidic linkages in the carbohydrate chain were confirmed by the ROESY and HMBC spectra of 7 (Table 1). The ROE correlation H(16)/H(18) observed in the spectrum of **7** and the closeness of the coupling constants of the protons H(15α), H(15β), H(16) and H(17) to those in kuriloside A, isolated earlier, confirmed the 16*α*-*O*Ac orientation, which was earlier established based on the different decoupling experiments performed with the protons of ring D followed by the comparison of the values of the experimental coupling constants of H(15α), H(15β), H(16) and H(17) in **7** with those calculated for the 16*α*- and 16*β*-substituted holostane derivatives [10].

The structure of kuriloside A (**7**) was also confirmed by the (–)HR-ESI-MS spectrum, in which the [M_Na_–Na]^−^ ion peak at an *m/z* of 1273.5196 (calc. 1273.5165) was observed, which corresponded to the molecular formula of C_56_H_89_O_30_SNa.

As result of our investigation, six unknown earlier triterpene glycosides were isolated from the sea cucumber *Thyonidium (= Duasmodactyla) kurilensis*. The glycosides have five different carbohydrate chains (kurilosides of the groups A, C–F), including three novel ones. The sulfated hexasaccharide moieties of kurilosides D (**4**) and F (**6**) are the third and fourth findings, correspondingly, in additional to the carbohydrate chains of the cladolosides of the groups K and L [11] of such type sugar parts in the sea cucumber glycosides. They all differ from each other in monosaccharide composition and the sulfate position. Pentasaccharide, branched by C(4)Xyl1, the chain of kuriloside E (**5**) with glucose as the second unit, is also unique. The oligosaccharide chains of the kurilosides of groups A and C, characterized by the same position of branching, were also found only in the glycosides from *T. kurilensis*. Five non-holostane aglycones without a lactone and with a 9(11) double bond were discovered in glycosides **1**–**7**. Four of them have shortened side chains (22,23,24,25,26,27-hexa-*nor*-lanostane aglycones) and one aglycone (in kuriloside D (**4**)) was characterized by a normal side chain and has never been found earlier. It should also be noted that only in the glycosides from *T. kurilensis* were the substituents at C(16) with an *α*-orientation found, while all the other glycosides with a 16-*O*-acetic group were characterized by their *β*-orientation [14,15].

### 2.2. Bioactivity of the Glycosides

The cytotoxic activities of compounds **1**–**7** against mouse neuroblastoma Neuro 2a, normal epithelial JB-6 cells and erythrocytes were studied (Table 12). Kuriloside A_1_ (**1**) was the most active compound in the series, demonstrating strong cytotoxicity against erythrocytes and JB-6 cells and a moderate effect against Neuro 2a cells. While kurilosides A_2_ (**2**) and A (**7**) were highly or moderately cytotoxic, respectively, against the erythrocytes and JB-6 cells, they were not effective against Neuro 2a cells, showing the influence of the aglycone structures on the activity of the glycosides. The presence of an *O*-acetic group at C(20) in **1** is apparently compensating for the absence of a normal side chain, resulting in its increasing cytotoxicity. The activity of kuriloside C_1_ (**3**) was decreased in relation of all the tested cell lines in comparison with **1** due to the lack of a glucose unit attached to C(4)Qui2. Kuriloside E (**5**) was the less active compound in the series due to the presence of a glucose residue as the second unit of the carbohydrate chain. Hexaoside with a non-methylated terminal glucose residue, kuriloside D (**4**), demonstrated a stronger activity against erythrocytes and JB-6 cells when compared with kuriloside F (**6**), which has a hexasaccharide chain with methylated terminal sugar units. This influence can be explained also by the presence of a normal non-shortened side chain in the aglycone of **4**. It is interesting that the glycosides with branched pentasaccharide chains (the kurilosides of group A) possessed higher cytotoxicity than those with hexasaccharide chains (the kurilosides of groups D and F).

## 3. Materials and Methods

### 3.1. General Experimental Procedures

We used for specific rotation a Perkin-Elmer 343 Polarimeter (Perkin-Elmer, Waltham, MA, USA); for NMR, a Bruker Avance III 700 Bruker FT-NMR (Bruker BioSpin GmbH, Rheinstetten, Germany) (700.00/176.03 MHz) (^1^H/^13^C) spectrometer; for ESI MS (positive and negative ion modes), an Agilent 6510 Q-TOF apparatus (Agilent Technology, Santa Clara, CA, USA), sample concentration of 0.01 mg/mL; and for HPLC, an Agilent 1100 apparatus with a differential refractometer (Agilent Technology, Santa Clara, CA, USA). The column was a Supelco Discovery HS F5-5 (10 × 250 mm, 5 μm) (Supelco, inc., Bellefonte, PA, USA).

### 3.2. Animals and Cells

Specimens of the sea cucumber *Thyonidium* (*= Duasmodactyla*) *kurilensis* (Levin) (family Cucumariidae; order Dendrochirotida) were collected in August of 1990 using an industrial rake-type dredge in the waters of the Onekotan Island (Kurile islands, the Sea of Okhotsk) at a depth of 100 m by a middle fishing trawler “Breeze” with a rear scheme of trawling during scallop harvesting. The sea cucumbers were identified by Prof. V.S. Levin; voucher specimens are preserved in the A.V. Zhirmunsky National Scientific Center of Marine Biology, Vladivostok, Russia.

CD-1 mice, weighing 18–20 g, were purchased from the RAMS “Stolbovaya” nursery (Stolbovaya, Moscow District, Russia) and kept at the animal facility in standard conditions.

All experiments were performed following the protocol for animal study approved by the Ethics Committee of the Pacific Institute of Bioorganic Chemistry No. 0085.19.10.2020. All experiments were also conducted in compliance with all of the rules and international recommendations of the European Convention for the Protection of Vertebrate Animals Used for Experimental Studies.

Mouse epithelial JB-6 cells Cl 41-5a and mouse neuroblastoma cell line Neuro 2a (ATCC^®^ CCL-131) were purchased from ATCC (Manassas, VA, USA).

### 3.3. Extraction and Isolation

The collected sea cucumbers were fixed by EtOH and then extracted twice with refluxing 60% EtOH. The extracts were evaporated to dryness and dissolved in water followed by chromatography on a Polychrom-1 column (powdered Teflon, Biolar, Latvia). The glycosides were eluted with 50% EtOH, and the fractions combined and evaporated. The first attempt to isolate the glycosides from another part of the sum was made in the early 1990s to obtain kurilosides A and C [10]. The remaining part of the crude glycosidic sum of *T. kurilensis* was stored at −18 °C. Then, it was separated by repeated chromatography on Si gel columns using CHCl_3_/EtOH/H_2_O (100:100:17) and (100:125:25) as mobile phases to give five fractions (I–V). Fraction I was subsequently subjected to HPLC on a reversed-phase semipreparative Supelco Discovery HS F5-5 (10 × 250 mm) column with MeOH/H_2_O/NH_4_OAc (1 M water solution) (70/29/1) as the mobile phase, resulting in the isolation of four subfractions (1–4) and an individual kuriloside C_1_ (**3**) (2.6 mg). Each of the subfractions 1–4 was submitted to rechromatography on the same column but with different ratios of MeOH/H_2_O/NH_4_OAc (1 M water solution) applied as the mobile phase. The use of the ratio (67/32/1) for subfraction 4 gave 12.6 mg of kuriloside A_1_ (**1**); (61/38/1) applied for subfraction 3 gave 13 mg of kuriloside D (**4**) and 3.3 mg of kuriloside E (**5**); (60/39/1) applied for subfraction 2 gave 17 mg of kuriloside A_2_ (**2**) and the same ratio used for the HPLC of subfraction 1 gave 42.4 mg of the known kuriloside A (**7**), as well as 4.5 mg of kuriloside F (**6**).

#### 3.3.1. Kuriloside A_1_ (**1**)

Colorless powder; [α]_D_^20^–5 (*c* 0.1, 50% MeOH). NMR: See Table 1 and Table 2, Appendix A. (–)HR-ESI-MS *m/z*: 1317.5449 (calc. 1317.5427) [M_Na_–Na]^−^; (–)ESI-MS/MS *m/z*: 1257.5 [M_Na_–Na–CH_3_COOH]^−^, 1197.5 [M_Na_–Na–2CH_3_COOH]^−^, 1035.4 [M_Na_–Na–2CH_3_COOH–C_6_H_10_O_5_ (Glc)]^−^, 889.4 [M_Na_–Na–2CH_3_COOH–C_6_H_10_O_5_ (Glc)–C_6_H_10_O_4_ (Qui)]^−^, 565.1 [M_Na_–Na–C_28_H_43_O_4_ (Agl)–C_6_H_10_O_5_ (Glc)–C_6_H_10_O_4_ (Qui)–H]^−^.

#### 3.3.2. Kuriloside A_2_ (**2**)

Colorless powder; [α]_D_^20^–4 (*c* 0.1, 50% MeOH). NMR: See Table 1 and Table 3, Appendix A. (–)HR-ESI-MS *m/z*: 1213.4964 (calc. 1213.4954) [M_Na_–Na]^−^; (–)ESI-MS/MS *m/z*: 1037.4 [M_Na_–Na–C_7_H_12_O_5_ (MeGlc)]^−^, 905.4 [M_Na_–Na–C_6_H_10_O_5_ (Glc)–C_6_H_10_O_4_ (Qui)]^−^, 565.1 [M_Na_–Na–C_24_H_35_O (Agl)–C_6_H_10_O_5_ (Glc)–C_6_H_10_O_4_ (Qui)–H]^−^.

#### 3.3.3. Kuriloside C_1_ (**3**)

Colorless powder; [α]_D_^20^–4 (*c* 0.1, 50% MeOH). NMR: See Table 2 and Table 4, Appendix A. (–)HR-ESI-MS *m/z*: 1155.4923 (calc. 1155.4899) [M_Na_–Na]^−^; (–)ESI-MS/MS *m/z*: 1095.5 [M_Na_–Na–CH_3_COOH]^−^, 1035.5 [M_Na_–Na–2CH_3_COOH]^−^, 889.4 [M_Na_–Na–2CH_3_COOH–C_6_H_10_O_4_ (Qui)]^−^, 565.1 [M_Na_–Na–C_28_H_43_O_4_ (Agl)–C_6_H_10_O_4_ (Qui)–H]^−^.

#### 3.3.4. Kuriloside D (**4**)

Colorless powder; [α]_D_^20^–14 (*c* 0.1, 50% MeOH). NMR: See Table 5 and Table 6, Appendix A. (–)HR-ESI-MS *m/z*: 1489.6174 (calc. 1489.6163) [M_Na_–Na]^−^; (–)ESI-MS/MS *m/z*: 1349.5 [M_Na_–Na–C_8_H_13_O_2_+H]^−^, 1187.5 [M_Na_–Na–C_8_H_13_O_2_–C_6_H_10_O_5_ (Glc)]^−^, 1025.4 [M_Na_–Na–C_8_H_13_O_2_–2C_6_H_10_O_5_ (Glc)]^−^, 879.4 [M_Na_–Na–C_8_H_13_O_2_–2C_6_H_10_O_5_ (Glc)–C_6_H_10_O_4_ (Qui)]^−^, 565.1 [M_Na_–Na–C_30_H_45_O_3_ (Agl)–2C_6_H_10_O_5_ (Glc)–C_6_H_10_O_4_ (Qui)–H]^−^.

#### 3.3.5. Kuriloside E (**5**)

Colorless powder; [α]_D_^20^–5 (*c* 0.1, 50% MeOH). NMR: See Table 7 and Table 8, Appendix A. (–)HR-ESI-MS *m/z*: 1231.5082 (calc. 1231.5059) [M_Na_–Na]^−^; (–)ESI-MS/MS *m/z*: 1069.5 [M_Na_–Na–C_6_H_10_O_5_ (Glc)]^−^, 1055.4 [M_Na_–Na–C_7_H_12_O_5_ (MeGlc)]^−^, 907.4 [M_Na_–Na–2C_6_H_10_O_5_ (Glc)]^−^, 565.1 [M_Na_–Na–C_24_H_37_O (Agl)–2C_6_H_10_O_5_ (Glc)–H]^−^.

#### 3.3.6. Kuriloside F (**6**)

Colorless powder; [α]_D_^20^–1 (*c* 0.1, 50% MeOH). NMR: See Table 9 and Table 10, Appendix A. (–)HR-ESI-MS *m/z*: 1407.5778 (calc. 1407.5744) [M_Na_–Na]^−^; (–)ESI-MS/MS *m/z*: 1231.5 [M_Na_–Na–C_7_H_12_O_5_ (MeGlc)]^−^, 1069.4 [M_Na_–Na–C_7_H_12_O_5_ (MeGlc)–C_6_H_10_O_5_ (Glc)]^−^, 923.4 [M_Na_–Na–C_7_H_12_O_5_ (MeGlc)–C_6_H_10_O_5_ (Glc)–C_6_H_10_O_4_ (Qui)]^−^, 565.1 [M_Na_–Na–C_24_H_37_O_2_ (Agl)–C_7_H_12_O_5_ (MeGlc)–C_6_H_10_O_5_ (Glc)–C_6_H_10_O_4_ (Qui)–H]^−^.

#### 3.3.7. Kuriloside A (**7**)

Colorless powder; See Table 1 and Table 10, Appendix A. (–)HR-ESI-MS *m/z*: 1273.5196 (calc. 1273.5165) [M_Na_–Na]^−^.

### 3.4. Cytotoxic Activity (MTT Assay)

All compounds were tested in concentrations from 1.5 μM to 100 μM using a two-fold dilution in dH_2_O. The solutions (20 µL) of the tested substances in different concentrations and a cell suspension (180 µL) were added in the wells of 96-well plates (1 × 10^4^ cells/well) and incubated for 24 h at 37 °C and at 5% CO_2_. After incubation, the medium with the tested substances was replaced by 100 μL of fresh medium. Then, 10 μL of an MTT (thiazoyl blue tertrazolium bromide) stock solution (5 mg/mL) was added to each well and the microplate was incubated for 4 h. After that, 100 μL of sodium dodecyl sulfate (SDS)-HCl solution (1 g SDS/10 mL dH_2_O/17 μL 6 N HCl) was added to each well followed by incubation for 18 h. The absorbance of the converted dye formazan was measured using a Multiskan FC microplate photometer (Thermo Fisher Scientific, Waltham, MA, USA) at a wavelength of 570 nm. The cytotoxic activity of the substances was calculated as the concentration that caused 50% metabolic cell activity inhibition (IC_50_). All the experiments were made in triplicate, with a *p* < 0.01 indicating a significant difference.

### 3.5. Hemolytic Activity

Blood was taken from CD-1 mice (18–20 g). Erythrocytes were isolated from the blood of albino CD-1 mice by centrifugation with phosphate-buffered saline (pH 7.4) for 5 min at 4 °C and at 450× *g* (LABOFUGE 400R, Heraeus, Hanau, Germany), repeated three times. Then the residue of the erythrocytes was resuspended in an ice-cold phosphate saline buffer (pH 7.4) to a final optical density of 1.5 at 700 nm and kept on ice [16]. For the hemolytic assay, 180 µL of erythrocyte suspension was mixed with 20 µL of the test compound solution in V-bottom 96-well plates. After 1 h of incubation at 37 °C, the plates were exposed to centrifugation for 10 min at 900× *g* using a laboratory centrifuge (LMC-3000, Biosan, Riga, Latvia) [16]. Then, we carefully selected 100 µL of the supernatant and transferred it into new flat-plates, respectively. Lysis of erythrocytes was determined by measuring the concentration of hemoglobin in the supernatant with a microplate photometer (Multiskan FC, Themo Fisher Scientific, Waltham, MA, USA), with λ = 570 nm [17]. The effective dose causing 50% hemolysis of the erythrocytes (ED_50_) was calculated using the computer program SigmaPlot 10.0. All the experiments were made in triplicate, with *p* < 0.01 indicating a significant difference.

## 4. Conclusions

It is known that triterpene glycosides of the sea cucumbers are formed by the mosaic type of biosynthesis [3,7]. From this viewpoint, the carbohydrate chains are biosynthesized independently from the aglycones by the stepwise glycosylation of the forming chain by individual monosaccharides, which bonded to certain positions only. Hence, the direction of the biosynthetic transformation of the sugar chains of compounds **1**–**7** is supposed to be as follows: the tetrasaccharide chain of the kurilosides of group C (known as kuriloside C [10] and C_1_ (**3**)) is a precursor for the pentasaccharide chains of the kurilosides of group A (**1**, **2**, **7**); the subsequent glycosylation leads to the formation of a hexasaccharide chain of kuriloside D (**4**) and further attachment of the O-methyl group to C(3) of the terminal (fourth) residue, resulting in the formation of the chain of kuriloside F (**6**). The carbohydrate chain of kuriloside E (**5**) is obviously branching from the mainstream biosynthesis because the C(2)-position of the first (xylose) residue is glycosylated by the glucose residue instead of the quinovose residue, common for this position of kurilosides and attributed to the other groups. The rest of the monosaccharide units in the chain of **5** are the same as in the kurilosides of group A (**1**, **2**, **7**).

The biogenetic relationships of the aglycone parts of compounds **1**–**7** are presented in Figure 2. The precursor of the *nor*-lanostane aglycones is the derivative that has a normal side chain, with an oxygen-containing substituent at C(22), which is a necessary condition for the subsequent bond cleavage between 20 and 22 carbons with elimination of a side chain portion that leads to the formation of the 22,23,24,25,26,27-hexa-*nor*-lanostane (4,4,14-trimethyl-pregnane) aglycones without a lactone like in all glycosides of *T. kurilensis*, except for kuriloside D (**4**). In the process of forming of all the other aglycones of the kurilosides, side-chain cleavage occurs. The aglycone of kuriloside E (**5**) corresponds to this stage of biosynthesis. The introduction of an α-hydroxyl group to C(16) leads to the aglycone of kuriloside F (**6**). The introduction of a β-hydroxyl group to C(16) also occurs, but it quickly transformed to acetylated derivatives. The intermolecular dehydration of the 16-hydroxylated precursors leads to the aglycone of kuriloside A_2_ (**2**). The acetylation of the 16α-hydroxy-group resulted in the synthesis of the aglycone of kuriloside A (**7**). The enzymatic 16β-*O*-acetylation followed by the reduction of the 20-keto group, with subsequent acetylation of this position, leads to the aglycones of kurilosides A_1_ (**1**) and C_1_ (**3**).

## Figures and Tables

**Figure 1 marinedrugs-18-00551-f001:**
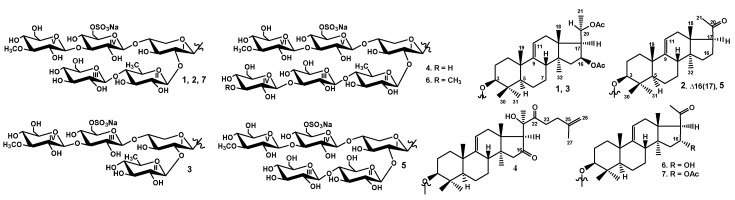
Chemical structures of the glycosides isolated from *Thyonidium kurilensis:*
**1**—kuriloside A_1_; **2**—kuriloside A_2_; **3**—kuriloside C_1_; **4**—kuriloside D; **5**—kuriloside E; **6**—kuriloside F; **7**—kuriloside A.

**Figure 2 marinedrugs-18-00551-f002:**
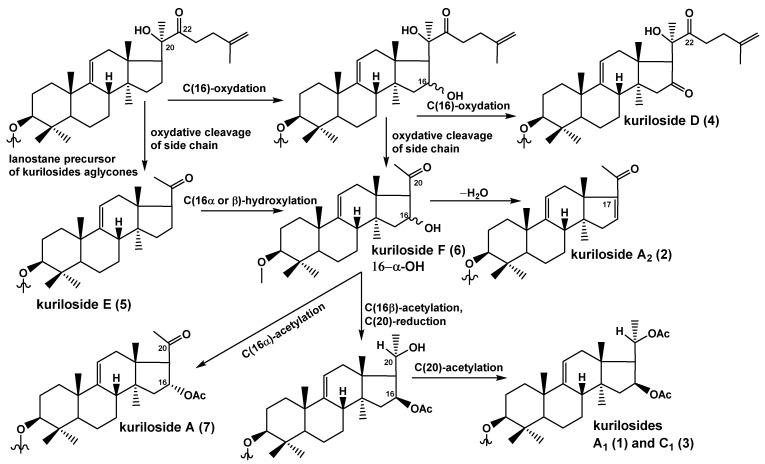
The biosynthetic pathways of the aglycones of the glycosides from *T. kurilensis.*

**Table 1 marinedrugs-18-00551-t001:** ^13^C and ^1^H NMR chemical shifts and HMBC and ROESY correlations of the carbohydrate moiety of kurilosides A_1_ (**1**), A_2_ (**2**) and A (**7**).

Atom.	δ_C_ mult. *^a,b,c^*	δ_H_ mult. *^d^* (*J* in Hz)	HMBC	ROESY
Xyl1 (1→C-3)				
1	105.0 CH	4.69 d (7.2)	C-3; C: 5 Xyl1	H-3; H-3, 5 Xyl1
2	**83.2** CH *^b^*	3.95 t (8.8)	C: 3 Xyl1	H-1 Qui2
3	75.7 CH	4.13 t (9.0)	C: 2, 4 Xyl1	
4	**79.8** CH *^b^*	4.07m		H-1 Glc4
5	63.8 CH_2_	4.29 dd (5.3; 11.3)	C: 1, 3 Xyl1	
		3.58 t (11.3)	C: 1 Xyl1	H-1 Xyl1
Qui2 (1→2Xyl1)				
1	105.3CH	5.02 d (7.6)	C: 2 Xyl1	H-2 Xyl1; H-3, 5 Qui2
2	76.1 CH	3.98 t (8.3)	C: 1, 3 Qui2	
3	75.3 CH	4.13 t (8.3)		
4	**86.8** CH *^b^*	3.57 t (8.3)	C: 1 Glc3; C: 3, 5 Qui2	H-1 Glc3
5	71.7 CH	3.76 dd (6.2; 8.3)		H-1 Qui2
6	17.9 CH_3_	1.70 d (6.2)	C: 4, 5 Qui2	
Glc3 (1→4Qui2)				
1	105.3 CH	4.92 d (8.0)	C: 4 Qui2	H-4 Qui2; H-3,5 Glc3
2	74.7 CH	4.00 t (8.5)	C: 1, 3 Glc3	
3	78.2 CH	4.21 t (9.1)	C: 2, 4Glc3	H-1 Glc3
4	71.4 CH	4.11 t (8.5)	C: 5, 6 Glc3	
5	78.4CH	4.05 m		H-1 Glc3
6	62.3 CH_2_	4.55 brd (10.7)		
		4.23 dd (6.4; 11.7)		
Glc4 (1→4Xyl1)				
1	103.7CH	4.87 d (8.0)	C: 4Xyl1	H-4 Xyl1; H-3, 5 Glc4
2	73.2 CH	3.89 t (8.9)	C: 1, 3 Glc4	
3	**86.9** CH *^b^*	4.16 t (8.9)	C: 2, 4 Glc4, C: 1 MeGlc5	H-1 MeGlc5; H-1 Glc4
4	69.6 CH	3.90 t (8.9)	C: 3, 5, 6 Glc4	
5	75.7 CH	4.13 m		H-1 Glc4
6	*67.3* CH_2_ *^c^*	5.18 d (9.8)		
		4.75 dd (6,3; 11.6)	C: 5 Glc4	
MeGlc5 (1→3Glc4)				
1	105.3 CH	5.25 d (8.7)	C: 3 Glc4	H-3 Glc4; H-3, 5 MeGlc5
2	74.9 CH	3.95 t (8.7)	C: 1, 3 MeGlc5	
3	87.8 CH	3.68 t (8.7)	C: 2, 4 MeGlc5, OMe	H-1 MeGlc5
4	70.3 CH	4.13 t (8.7)	C: 3, 5, 6 MeGlc5	
5	78.1 CH	3.93 m		H-1, 3 MeGlc5
6	61.9 CH_2_	4.44 d (12.0)	C: 4 MeGlc5	
		4.26 dd (5.5; 12.0)		
OMe	60.5 CH_3_	3.85 s	C: 3 MeGlc5	

*^a^* Recorded at 176.03 MHz in C_5_D_5_N/D_2_O (4/1). *^b^* Bold = interglycosidic positions. *^c^* Italic = sulphate position. *^d^* Recorded at 700.00 MHz in C_5_D_5_N/D_2_O (4/1). Multiplicity by 1D TOCSY.

**Table 2 marinedrugs-18-00551-t002:** ^13^C and ^1^H NMR chemical shifts and HMBC and ROESY correlations of the aglycone moiety of kurilosides A_1_ (**1**) and C_1_ (**3**).

Position	δ_C_ mult. *^a^*	δ_H_ mult. (*J* in Hz) *^b^*	HMBC	ROESY
1	36.2 CH_2_	1.77 m		H-11, H-19
		1.38 m		H-3, H-5, H-11
2	27.0 CH_2_	2.19 m		H-19
		1.94 m		H-19, H-30
3	88.3 CH	3.18 dd (4.2; 11.9)	C: 1, 30, 31, C-1Xyl1	H-1, H-5, H-31, H-1Xyl1
4	39.7 C			
5	52.7 CH	0.87 brd (12.3)	C: 6, 10, 19, 30	H-1, H-3, H-7, H-31
6	21.1 CH_2_	1.70 m		H-30, H-31
		1.50 m		H-19, H-30
7	28.0 CH_2_	1.58 m		H-15
		1.27 m		H-5, H-32
8	41.2 CH	2.20 m		H-18
9	148.9 C			
10	39.3 C			
11	114.0 CH	5.23 brd (5.6)	C: 8, 10, 12, 14	H-1
12	35.8 CH_2_	2.08 m		H-17, H-32
		1.78 m	C: 9, 11, 13, 15, 18	H-18, H-21
13	45.5 C			
14	43.6 C			
15	43.7 CH_2_	2.10 dd (6.3; 13.6)	C: 32	H-32
		1.35 dd (5.0; 13.6)	C: 8, 13, 16, 32	H-18
16	73.8 CH	5.64 dd (5.2; 7.7; 13.4)	C: 15; OAc-16	H-32; OAc-16
17	53.3 CH	2.41 dd (7.7; 10.6)	C: 12, 15, 18, 20, 21	H-12, H-21, H-32
18	15.2 CH_3_	0.82 s	C: 12, 13, 14, 17	H-8, H-12, H-19, H-20, H-21
19	22.3 CH_3_	1.14 s	C: 1, 5, 9, 10	H-1, H-2, H-6, H-8, H-18, H-30
20	69.4 CH	5.46 dd (6.1; 10.6)	C: 16, 17, 21, OAc-20	H-21, OAc-20
21	19.6 CH_3_	1.32 d (6.0)	C: 17, 20	H-12, H-17, H-18, H-20
30	16.4 CH_3_	1.04 s	C: 3, 4, 5, 31	H-2, H-6, H-31
31	27.8 CH_3_	1.24 s	C: 3, 4, 5, 30	H-3, H-5, H-6, H-30, H-1 Xyl1
32	18.9 CH_3_	0.76 s	C: 8, 13, 14, 15	H-7, H-12, H-15, H-16, H-17
**C**OOCH_3_-16	169.8 C			
COO**C**H_3_-16	20.2 CH_3_	2.12 s		H-16, H-18
**C**OOCH_3_-20	169.9 C			
COO**C**H_3_-20	21.0 CH_3_	2.05 s		H-21

*^a^* Recorded at 176.03 MHz in C_5_D_5_N/D_2_O (4/1). *^b^* Recorded at 700.00 MHz in C_5_D_5_N/D_2_O (4/1).

**Table 3 marinedrugs-18-00551-t003:** ^13^C and ^1^H NMR chemical shifts and HMBC and ROESY correlations of the aglycone moiety of kuriloside A_2_ (**2**).

Position	δ_C_ mult. *^a^*	δ_H_ mult. (*J* in Hz) *^b^*	HMBC	ROESY
1	36.3 CH_2_	1.71 m		H-11
		1.35 m		H-11
2	27.0 CH_2_	2.15 m		
		1.89 m		H-19
3	88.5 CH	3.17 dd (4.2; 11.9)	C: 4, 30, 31, C:1 Xyl1	H-1, H-5, H-31, H-1Xyl1
4	39.7 C			
5	52.9 CH	0.90brd (1.8; 11.9)	C: 4, 6, 10, 19, 30, 31	H-1, H-3, H-31
6	21.1 CH_2_	1.71 m	C: 8	H-31
		1.49 m		H-30
7	27.9 CH_2_	1.65 m		
		1.31 m		H-32
8	39.4 CH	2.38 brd (9.6)		H-15, H-18, H-19
9	149.1 C			
10	39.4 C			
11	115.3 CH	5.33 brd (6.0)	C: 10, 12, 13	H-1
12	32.2 CH_2_	2.58 dd (5.7; 16.6)	C: 9, 11, 13, 14, 18	H-18
		2.47 brdd (2.7; 16.6)	C: 9, 11, 18	H-32
13	49.7 C			
14	47.1 C			
15	41.7 CH_2_	2.24 d (16.6)	C: 8, 14, 16, 17, 32	H-18
		2.05dd (3.0; 16.6)	C: 13, 14, 16, 17, 32	H-32
16	144.5 CH	6.63 brt (2.6)	C: 13, 14, 15, 17, 20	H-21
17	152.1 C			
18	19.4 CH_3_	0.98 s	C: 12, 13, 14, 17	H-8, H-12, H-15
19	22.2 CH_3_	1.10 s	C: 1, 5, 9, 10	H-1, H-2, H-6, H-8
20	196.3 C			
21	26.8 CH_3_	2.28 s	C: 16, 17, 20	H-16
30	16.5 CH_3_	1.06 s	C: 3, 4, 5, 31	H-2, H-6, H-31
31	27.9 CH_3_	1.24 s	C: 3, 4, 5, 30	H-3, H-5, H-6, H-30, H-1 Xyl1
32	19.8 CH_3_	0.86 s	C: 8, 13, 14, 15	H-7, H-12, H-15

*^a^* Recorded at 176.03 MHz in C_5_D_5_N/D_2_O (4/1). *^b^* Recorded at 700.00 MHz in C_5_D_5_N/D_2_O (4/1).

**Table 4 marinedrugs-18-00551-t004:** ^13^C and ^1^H NMR chemical shifts and HMBC and ROESY correlations of carbohydrate moiety of kuriloside C_1_ (**3**).

Atom	δ_C_ mult. *^a,b,c^*	δ_H_ mult. (*J* in Hz) *^d^*	HMBC	ROESY
Xyl1 (1→C-3)				
1	104.9 CH	4.66 d (6.9)	C: 3	H-3; H-3, 5 Xyl1
2	**81.8** CH *^b^*	3.95t (7.5)	C: 1, 3 Xyl1	H-1 Qui2
3	74.8 CH	4.14 t (7.5)	C: 4 Xyl1	H-1, 5 Xyl1
4	**78.1** CH *^b^*	4.14 m		H-1 Glc3
5	63.5 CH_2_	4.37 dd (3.8; 10.7)	C: 3 Xyl1	H-3 Xyl1
		3.64 t (10.7)		H-1, 3 Xyl1
Qui2 (1→2Xyl1)				
1	104.8CH	5.01 d (7.8)	C: 2 Xyl1	H-2 Xyl1; H-5 Qui2
2	76.2 CH	3.80 t (8.6)	C: 1 Qui2	
3	76.7 CH	3.99 t (8.6)	C: 2, 4 Qui2	H-1, 5 Qui2
4	76.2 CH	3.52 t (8.6)	C: 3, 5 Qui2	H-2, 6 Qui2
5	72.9 CH	3.67 dd (6.2; 8.6)		H-1 Qui2
6	18.3 CH_3_	1.49 d (6.2)	C: 4, 5 Qui2	H-4 Qui2
Glc3 (1→4Xyl1)				
1	102.3 CH	4.88 d (8.2)	C: 4 Xyl1	H-4 Xyl1; H-3,5 Glc3
2	73.4 CH	3.83 t (9.0)	C: 1, 3 Glc3	
3	**85.9** CH *^b^*	4.13 t (9.0)	C: 1 MeGlc4; C: 2, 4Glc3	H-1 MeGlc4
4	69.1 CH	3.87 t (9.0)	C: 3, 5, 6 Glc3	H-6 Glc3
5	75.0CH	4.01 t (9.0)		H-1 Glc3
6	*67.5* CH_2_ *^c^*	4.83 d (11.5)		
		4.61 dd (5.7; 11.5)		H-4 Glc3
MeGlc4 (1→3Glc3)				
1	104.3 CH	5.13 d (8.0)	C: 3 Glc3	H-3 Glc3; H-3, 5 MeGlc4
2	74.5 CH	3.78 t (8.8)	C: 1 MeGlc4	
3	86.7 CH	3.63 t (8.8)	C: 2, 4 MeGlc4, OMe	H-1 MeGlc4
4	70.3 CH	3.83 t (8.8)	C: 5 MeGlc4	
5	77.4 CH	3.83 m		H-1 MeGlc4
6	61.8 CH_2_	4.27 d (12.0)		H-4 MeGlc4
		3.98 dd (6.4; 12.0)	C: 5 MeGlc4	
OMe	60.8 CH_3_	3.79 s	C: 3 MeGlc4	

*^a^* Recorded at 176.03 MHz in C_5_D_5_N/D_2_O (4/1). *^b^* Bold = interglycosidic positions. *^c^* Italic = sulphate position. *^d^* Recorded at 700.00 MHz in C_5_D_5_N/D_2_O (4/1). Multiplicity by 1D TOCSY.

**Table 5 marinedrugs-18-00551-t005:** ^13^C and ^1^H NMR chemical shifts and HMBC and ROESY correlations of the carbohydrate moiety of kuriloside D (**4**).

Atom	δ_C_ mult. *^a,b,c^*	δ_H_ mult. (*J* in Hz) *^d^*	HMBC	ROESY
Xyl1 (1→C-3)				
1	105.0 CH	4.70 d (7.6)	C: 3	H-3; H-3, 5 Xyl1
2	**83.3** CH *^b^*	3.95 t (8.3)	C: 3 Xyl1	H-1 Qui2
3	75.4 CH	4.14 t (8.3)	C: 2 Xyl1	H-1 Xyl1
4	**79.7** CH *^b^*	4.07 m		H-1 Glc5
5	63.8 CH_2_	4.30 dd (6.1; 12.1)		
		3.59 dd (9.1; 12.1)		H-1 Xyl1
Qui2 (1→2Xyl1)				
1	105.3 CH	5.02 d (7.6)	C: 2 Xyl1	H-2 Xyl1; H-3, 5 Qui2
2	75.8 CH	3.95 t (8.3)		H-4 Qui2
3	75.3 CH	4.05 t (8.3)	C: 2, 4 Qui2	H-1 Qui2
4	**87.1** CH *^b^*	3.56 t (8.3)	C: 1 Glc3; C: 5 Qui2	H-3 Glc3; H-2 Qui2
5	71.6 CH	3.76 dd (6.1; 9.7)		H-1, 3 Qui2
6	17.9 CH_3_	1.71 d (6.1)	C: 4, 5 Qui2	
Glc3 (1→4Qui2)				
1	104.8 CH	4.90 d (7.5)	C: 4 Qui2	H-4 Qui2
2	73.6 CH	4.03 t (8.2)	C: 1, 3 Glc3	
3	**88.1** CH *^b^*	4.20 m	C: 4 Glc3	H-1 Glc4; H-1 Glc3
4	69.7 CH	4.00 m	C: 3, 5 Glc3	
5	78.5 CH	4.00 m		
6	62.1 CH_2_	4.47 d (12.3)		
		4.13 m		
Glc4 (1→3Glc3)				
1	105.7 CH	5.28 d (8.2)	C: 3 Glc3	H-3 Glc3; H-3, 5 Glc4
2	75.3 CH	4.06 t (9.1)	C: 1, 3 Glc4	
3	77.9 CH	4.22 t (9.1)	C: 2, 4 Glc4	
4	71.5 CH	4.15 t (9.1)	C: 5, 6 Glc4	
5	78.1 CH	4.00 m		H-1 Glc4
6	62.4 CH_2_	4.51 dd (3.0; 11.5)		
		4.28 dd (5.4; 11.5)		
Glc5 (1→4Xyl1)				
1	103.7 CH	4.86 d (7.8)	C: 4 Xyl1	H-4 Xyl1; H-3 Glc5
2	73.2 CH	3.88 t (7.8)	C: 1, 3 Glc5	
3	**87.0** CH *^b^*	4.14 t (9.2)	C: 1 MeGlc6; C: 2, 4 Glc5	H-1 MeGlc6; H-1 Glc5
4	69.6 CH	3.92 t (9.2)	C: 3, 5, 6 Glc5	
5	76.1 CH	4.11 m		
6	*67.2* CH_2_ *^c^*	5.18 d (9.9)		
		4.76 dd (6.6; 11.2)	C: 5 Glc5	
MeGlc6 (1→3Glc4)				
1	105.3 CH	5.24 d (7.9)	C: 3 Glc5	H-3 Glc5; H-3, 5 MeGlc6
2	74.9 CH	3.95 t (8.6)	C: 1, 3 MeGlc6	
3	87.8 CH	3.68 t (8.6)	C: 2, 4 MeGlc6; OMe	H-1, 5 MeGlc6; OMe
4	70.4 CH	4.13 t (8.6)	C: 3, 6 MeGlc6	H-6 MeGlc6
5	78.2 CH	3.92 t (8.6)		H-1, 3 MeGlc6
6	61.9 CH_2_	4.44 dd (2.6; 11.8)	C: 4 MeGlc6	
		4.26 dd (5.3; 11.8)	C: 5 MeGlc6	
OMe	60.5 CH_3_	3.85 s	C: 3 MeGlc6	

*^a^* Recorded at 176.04 MHz in C_5_D_5_N/D_2_O (4/1). *^b^* Bold = interglycosidic positions. *^c^* Italic = sulfate position. *^d^* Recorded at 700.13 MHz in C_5_D_5_N/D_2_O (4/1). Multiplicity by 1D TOCSY.

**Table 6 marinedrugs-18-00551-t006:** ^13^C and ^1^H NMR chemical shifts and HMBC and ROESY correlations of the aglycone moiety of kuriloside D (**4**).

Position	δ_C_ mult. *^a^*	δ_H_ mult. (*J* in Hz) *^b^*	HMBC	ROESY
1	36.1 CH_2_	1.76 m		H-11, H-19
		1.39 m		H-3, H-11
2	26.9 CH_2_	2.19 m		
		1.92 m		H-19, H-30
3	88.4 CH	3.19 dd (4.0; 12.0)	C: 30, C: 1 Xyl1	H-1, H-5, H-31, H1-Xyl1
4	39.7 C			
5	52.7 CH	0.88 m		H-1, H-3, H-7
6	21.0 CH_2_	1.70 m		
		1.45 m		H-19
7	28.2 CH_2_	1.49 m		
		1.28 m		H-32
8	40.2 CH	2.33 m		H-18, H-19
9	149.0 C			
10	39.4 C			
11	115.0 CH	5.35 brd (6.2)	C: 10, 13	H-1
12	36.5 CH_2_	2.44 brd (16.4)		H-17, H-32
		2.20 dd (4.0; 16.8)	C: 9, 14	
13	43.7 C			
14	41.9 C			
15	48.1 CH_2_	2.27 d (17.7)	C: 14, 16, 32	H-18
		2.03 d (17.7)	C: 13, 16, 32	H-7, H-32
16	216.6 C			
17	63.9 CH	3.67 s	C: 12, 13, 16, 18, 20	H-12, H-21, H-32
18	16.9 CH_3_	1.28 s	C: 12, 13, 14, 17	H-8, H-19, H-21
19	22.2 CH_3_	1.11 s	C: 1, 5, 9, 10	H-1, H-2, H-6, H-8
20	80.9 C			
21	24.5 CH_3_	1.59 s	C: 17, 20, 22	H-12, H-17, H-18, H-23
22	216.5 C			
23	35.2 CH_2_	3.55 dd (5.8; 9.8)	C: 22	
		3.24 ddd (6.2; 9.3; 18.2)	C: 22, 24, 25	
24	31.8 CH_2_	2.59 m	C: 23, 25, 26, 27	
25	145.5 C			
26	110.0 CH_2_	4.87 brs	C: 24, 27	
		4.79 brs	C: 24, 27	
27	22.6 CH_3_	1.73 s	C: 24, 25, 26	H-23, H-24, H-26
30	16.5 CH_3_	1.06 s	C: 3, 4, 5, 31	H-2, H-6, H-31
31	27.9 CH_3_	1.25 s	C: 3, 4, 5, 30	H-3, H-5, H-6, H-30, H-1 Xyl1
32	18.7 CH_3_	0.90 s	C: 8, 13, 14, 15	H-7, H-12, H-15, H-17

*^a^* Recorded at 176.03 MHz in C_5_D_5_N/D_2_O (4/1). *^b^* Recorded at 700.00 MHz in C_5_D_5_N/D_2_O (4/1).

**Table 7 marinedrugs-18-00551-t007:** ^13^C and ^1^H NMR chemical shifts and HMBC and ROESY correlations of the carbohydrate moiety of kuriloside E (**5**).

Atom	δ_C_ mult. *^a,b,c^*	δ_H_ mult. (*J* in Hz) *^d^*	HMBC	ROESY
Xyl1 (1→C-3)				
1	105.1 CH	4.71 d (7.4)	C: 3	H-3; H-3, 5 Xyl1
2	**82.7** CH *^b^*	4.02t (9.7)	C: 1 Xyl1	H-1 Glc2
3	75.6 CH	4.16 t (9.7)	C: 4 Xyl1	H-1, 5 Xyl1
4	**80.8** CH *^b^*	4.09 m		H-1 Glc4
5	63.7 CH_2_	4.29 dd (5.9; 12.6)	C: 3 Xyl1	
		3.60 dd (9.7; 12.2)		H-1, 3 Xyl1
Glc2 (1→2Xyl1)				
1	105.4CH	5.10 d (7.7)	C: 2 Xyl1	H-2 Xyl1; H-3, 5 Glc2
2	76.1 CH	4.00 t (8.6)		
3	75.2 CH	4.30 t (8.6)		H-1, 5 Glc2
4	**81.0** CH *^b^*	4.30 t (8.6)	C: 1 Glc3	H-1 Glc3
5	76.5 CH	3.85 m		H-1, 3 Glc2
6	61.5 CH_2_	4.55 brd (12.2)		
		4.34 dd (3.2; 12.2)		
Glc3 (1→4Glc2)				
1	104.8 CH	5.11 d (7.0)	C: 4 Glc2	H-4 Glc2; H-3, 5 Glc3
2	74.6 CH	4.04 t (8.9)	C: 1, 3 Glc3	
3	77.9 CH	4.18 t (8.9)	C: 2, 4Glc3	H-1 MeGlc5; H-1, 5 Glc3
4	71.3 CH	4.12 t (8.9)	C: 5, 6 Glc3	
5	78.3CH	4.00 m		H-1, 3 Glc3
6	62.1 CH_2_	4.48 d (12.2)		
		4.22 dd (7.0; 12.2)		
Glc4 (1→4Xyl1)				
1	104.0 CH	4.90 d (7.8)	C: 4 Xyl1	H-4 Xyl1; H-3 Glc4
2	73.3 CH	3.94 t (8.6)	C: 1, 3 Glc4	
3	**87.0** CH *^b^*	4.18 t (8.6)	C: 1 MeGlc5; C: 2, 4 Glc4	H-1 Glc4
4	69.8 CH	3.88 t (8.6)		H-6 Glc4
5	75.7 CH	4.20 m		
6	*67.5* CH_2_ *^c^*	5.23 d (10.9)		
		4.72 t (10.1)		H-4 Glc4
MeGlc5 (1→3Glc4)				
1	105.4 CH	5.27 d (7.8)	C: 3 Glc4	H-3 Glc4; H-3, 5 MeGlc5
2	74.9 CH	3.96 t (8.6)	C: 1, 3 MeGlc5	H-4 MeGlc5
3	87.8 CH	3.69 t (8.6)	C: 2, 4 MeGlc5; OMe	H-1, 5 MeGlc5; OMe
4	70.3 CH	4.15 t (8.6)	C: 3, 5, 6MeGlc5	
5	78.2 CH	3.94 m		H-1, 3 MeGlc5
6	61.9 CH_2_	4.45 brd (9.4)		
		4.27 dd (5.5; 11.7)	C: 5 MeGlc5	
OMe	60.5 CH_3_	3.86 s	C: 3 MeGlc5	H-3 MeGlc5

*^a^* Recorded at 176.04 MHz in C_5_D_5_N/D_2_O (4/1). *^b^* Bold = interglycosidic positions. *^c^* Italic = sulfate position. *^d^* Recorded at 700.13 MHz in C_5_D_5_N/D_2_O (4/1). Multiplicity by 1D TOCSY.

**Table 8 marinedrugs-18-00551-t008:** ^13^C and ^1^H NMR chemical shifts and HMBC and ROESY correlations of the aglycone moiety of kuriloside E (**5**).

Position	δ_C_ mult. *^a^*	δ_H_ mult. (*J* in Hz) *^b^*	HMBC	ROESY
1	36.3 CH_2_	1.77 m		
		1.39 m		H-3, H-11
2	26.9 CH_2_	2.19 m		
		2.00 m		
3	88.4 CH	3.18 dd (4.1; 11.9)	C: 30, C:1 Xyl1	H-1, H-5, H-31, H-1Xyl1
4	39.6 C			
5	52.8 CH	0.85d (11.9)		H-1, H-3, H-7, H-31
6	21.1 CH_2_	1.61 m		H-31
		1.42 m		H-8
7	28.3 CH_2_	1.60 m		H-5
		1.26 m		H-32
8	41.4 CH	2.15m		H-6, H-18, H-19
9	149.1 C			
10	39.2 C			
11	114.1 CH	5.29 brd (6.2)	C: 10, 14	H-1
12	36.0 CH_2_	2.32 brdd (2.1; 16.2)		H-17, H-32
		1.96 dd (6.0; 16.2)	C: 9, 11, 13	H-18, H-21
13	47.5 C			
14	45.9 C			
15	33.9 CH_2_	1.43 m		H-18
		1.37 m		H-32
16	21.9 CH_2_	2.50 m		H-18
		1.68 m		H-32
17	59.7 CH	2.98t (9.0)	C: 14, 18, 20	H-12, H-21, H-32
18	16.4 CH_3_	0.63 s	C: 12, 13, 14, 17	H-8, H-12, H-15, H-16, H-19
19	22.3 CH_3_	1.10 s	C: 9	H-2, H-18
20	208.8 C			
21	30.7 CH_3_	2.12 s	C: 17, 20	H-12, H-17, H-18
30	16.5 CH_3_	1.05 s	C: 3, 4, 5, 31	H-2, H-6, H-31
31	27.9 CH_3_	1.18 s	C: 3, 4, 5, 30	H-3, H-5, H-6, H-30, H-1 Xyl1
32	18.6 CH_3_	0.81 s	C: 8, 13, 14, 15	H-7, H-12, H-15, H-16, H-17

*^a^* Recorded at 176.03 MHz in C_5_D_5_N/D_2_O (4/1). *^b^* Recorded at 700.00 MHz in C_5_D_5_N/D_2_O (4/1).

**Table 9 marinedrugs-18-00551-t009:** ^13^C and ^1^H NMR chemical shifts and HMBC and ROESY correlations of the carbohydrate moiety of kuriloside F (**6**).

Atom	δ_C_ mult. *^a,b,c^*	δ_H_ mult. (*J* in Hz) *^d^*	HMBC	ROESY
Xyl1 (1→C-3)				
1	104.7 CH	4.70 d (7.2)	C: 3	H-3; H-3, 5 Xyl1
2	**83.2** CH *^b^*	3.96 m	C: 1 Xyl1	H-1 Qui2
3	75.7 CH	4.13 t (8.8)	C: 2 Xyl1	
4	**79.8** CH *^b^*	4.07 m		H-1 Glc5
5	63.8 CH_2_	4.29 dd (5.3; 11.4)	C: 3 Xyl1	
		3.58 dd (9.7; 11.4)		H-1 Xyl1
Qui2 (1→2Xyl1)				
1	105.0CH	5.02 d (7.3)	C: 2 Xyl1	H-2 Xyl1; H-5 Qui2
2	76.0 CH	3.97 t (9.4)	C: 1 Qui2	
3	75.3 CH	4.10 m		
4	**86.9** CH *^b^*	3.56 t (9.4)	C: 1 Glc3; C: 3, 5 Qui2	H-1 Glc3
5	71.6 CH	3.76 m		H-1 Qui2
6	17.8 CH_3_	1.70 d (5.9)	C: 4, 5 Qui2	
Glc3 (1→4Qui2)				
1	104.7 CH	4.90 d (7.4)	C: 4 Qui2	H-4 Qui2; H-3, 5 Glc3
2	73.5 CH	4.01 t (8.3)	C: 1, 3 Glc3	
3	87.8 CH	4.20 t (8.3)	C: 4Glc3	H-1 MeGlc4; H-1, 5 Glc3
4	69.6 CH	4.00 m		
5	77.9CH	4.00 t (8.3)		H-1, 3 Glc3
6	61.9 CH_2_	4.45 m		
		4.14 dd (5.5; 12.0)		
MeGlc4 (1→3Glc3)				
1	105.5 CH	5.26 d (8.1)	C: 3 Glc3	H-3 Glc3; H-3, 5MeGlc4
2	74.9 CH	3.99 t (8.1)	C: 1, 3 MeGlc4	
3	87.8 CH	3.70 t (8.1)	C: 2, 4 MeGlc4; OMe	H-1 MeGlc4; OMe
4	70.5 CH	4.13 t (8.5)	C: 3, 5, 6 MeGlc4	
5	78.2 CH	3.95 m		H-1 MeGlc4
6	62.1 CH_2_	4.46 dd (4.1; 12.2)	C: 4 MeGlc4	
		4.26 dd (6.1; 11.2)	C: 5 MeGlc4	
OMe	60.5 CH_3_	3.85 s	C: 3 MeGlc4	H-3 MeGlc4
Glc5 (1→4Xyl1)				
1	103.7 CH	4.86 d (8.1)	C: 4 Xyl1	H-4 Xyl1; H-3 Glc5
2	73.2 CH	3.86 t (8.1)	C: 1 Glc5	
3	86.9 CH	4.15 t (9.1)	C: 1 MeGlc6; C: 2, 4 Glc5	H-1 MeGlc6
4	69.6 CH	3.91 m	C: 3 Glc5	
5	75.7 CH	4.13 m		H-1 Glc5
6	*67.2* CH_2_ *^c^*	5.19 d (11.2)		
		4.77 dd (5.1; 11.2)		
MeGlc6 (1→3Glc5)				
1	105.3 CH	5.25 d (8.6)	C: 3 Glc5	H-3 Glc5; H-3, 5 MeGlc6
2	74.8 CH	3.96 t (8.6)	C: 1, 3 MeGlc6	
3	87.9 CH	3.68 t (8.6)	C: 2, 4 MeGlc6; OMe	H-1, 5 MeGlc6; OMe
4	70.3 CH	4.14 t (8.6)	C: 3, 5, 6MeGlc6	
5	78.2 CH	3.93 m		H-1, 3 MeGlc6
6	61.9 CH_2_	4.44 brd (11.8)	C: 4 MeGlc6	
		4.27 dd (5.5; 11.8)	C: 5 MeGlc6	
OMe	60.6 CH_3_	3.86 s	C: 3 MeGlc6	H-3 MeGlc6

*^a^* Recorded at 176.04 MHz in C_5_D_5_N/D_2_O (4/1). *^b^* Bold = interglycosidic positions. *^c^* Italic = sulfate position. *^d^* Recorded at 700.13 MHz in C_5_D_5_N/D_2_O (4/1). Multiplicity by 1D TOCSY.

**Table 10 marinedrugs-18-00551-t010:** ^13^C and ^1^H NMR chemical shifts and HMBC and ROESY correlations of the aglycone moiety of kuriloside F (**6**).

Position	δ_C_ mult. *^a^*	δ_H_ mult. (*J* in Hz) *^b^*	HMBC	ROESY
1	36.2 CH_2_	1.77 m		H-11
		1.41 m		H-3, H-5, H-11, H-31
2	27.0 CH_2_	2.19 m		
		1.94 m		
3	88.4 CH	3.19 dd (4.6; 12.5)	C: 4, 30, 31, C:1 Xyl1	H-1, H-5, H-31, H1-Xyl1
4	39.7 C			
5	52.8 CH	0.92 brd (12.5)	C: 4, 10, 19, 30	H-1, H-3, H-7, H-31
6	21.2 CH_2_	1.70 m		H-31
		1.47 m		H-19
7	28.4 CH_2_	1.64 m		
		1.36 m		
8	41.5 CH	2.16 m		H-18, H-19
9	149.0 C			
10	39.4 C			
11	114.2 CH	5.31 brd (6.3)	C: 8, 10, 12, 13	H-1
12	35.8 CH_2_	2.48 brd (16.1)		H-17, H-32
		1.94 dd (6.3; 16.1)	C: 9, 11, 13, 18	H-8, H-18
13	46.9 C			
14	46.8 C			
15	45.2 CH_2_	2.07 dd (9.2; 13.2) β	C: 8, 13, 32	H-16, H-18
		1.79 d (13.2) α	C: 14, 16, 32	H-32
16	71.1 CH	5.40 brt (7.5)	C: 13, 14, 20	H-15β, H-18
17	70.0 CH	3.40 d (6.4)	C: 12, 13, 14, 16, 18, 20	H-12, H-21, H-32
18	17.3 CH_3_	0.71 s	C: 12, 13, 17	H-8, H-12, H-15β, H-16, H-19
19	22.3 CH_3_	1.10 s	C: 1, 5, 9, 10	H-1, H-2, H-6, H-8, H-18
20	208.8 C			
21	31.2 CH_3_	2.18 s	C: 17, 20	H-12, H-17
30	16.5 CH_3_	1.05 s	C: 3, 4, 5, 31	H-2, H-6, H-31, H-6 Qui2
31	27.9 CH_3_	1.24 s	C: 3, 4, 5, 30	H-3, H-5, H-6, H-30, H-1 Xyl1
32	20.0 CH_3_	1.24 s	C: 8, 13, 15	H-7, H-12, H-15, H-17

*^a^* Recorded at 176.03 MHz in C_5_D_5_N/D_2_O (4/1). *^b^* Recorded at 700.00 MHz in C_5_D_5_N/D_2_O (4/1).

**Table 11 marinedrugs-18-00551-t011:** ^13^C and ^1^H NMR chemical shifts and HMBC and ROESY correlations of the aglycone moiety of kuriloside A (**7**).

Position	δ_C_ mult. *^a^*	δ_H_ mult. (*J* in Hz) *^b^*	HMBC	ROESY
1	36.1 CH_2_	1.75 m		H-11, H-30
		1.40 m		H-5, H-11
2	27.0 CH_2_	2.18 m		
		1.90 m		H-30
3	88.4 CH	3.19 dd (4.2; 11.9)	C: 4, 30, 31, C:1 Xyl1	H-1, H-5, H-31, H1-Xyl1
4	39.7 C			
5	52.8 CH	0.90 brd (11.9)	C: 4, 10, 19, 30	H-1, H-3, H-7, H-31
6	21.2 CH_2_	1.68 m		H-31
		1.43 m		H-30
7	28.3 CH_2_	1.59 m		
		1.32 m		H-5, H-32
8	41.3 CH	2.12 m		H-18, H-19
9	149.0 C			
10	39.4 C			
11	114.1 CH	5.29 brd (5.6)	C: 8, 10, 12, 13	H-1
12	35.6 CH_2_	2.45 brd (17.3)	C: 9	H-17, H-21, H-32
		1.94 dd (6.5; 17.3)	C: 9, 11, 13	H-18
13	46.7 C			
14	46.2 C			
15	42.6 CH_2_	2.07 brdd (10.0; 14.7) β	C: 8, 13, 32	
		1.58 d (14.8) α	C: 14, 16, 17, 32	H-32
16	75.4 CH	6.03 brt (7.5)	C: 13, 17, 20, OAc	H-18
17	65.8 CH	3.39 d (6.2)	C: 12, 13, 14, 16, 18, 20	H-12, H-21, H-32
18	17.1 CH_3_	0.65 s	C: 12, 13, 17	H-8
19	22.2 CH_3_	1.05 s	C: 1, 5, 9, 10	H-1, H-2, H-6, H-8, H-18
20	206.7 C			
21	30.7 CH_3_	2.19 s	C: 17, 20	H-12, H-17, H-18
30	16.5 CH_3_	1.05 s	C: 3, 4, 5, 31	H-2, H-6, H-31
31	27.9 CH_3_	1.23 s	C: 3, 4, 5, 30	H-3, H-5, H-6, H-30, H-1 Xyl1
32	19.4 CH_3_	1.02 s	C: 8, 13, 15	H-7, H-12, H-17
**C**OOCH_3_	170.4 C			
OCO**C**H_3_	20.9 CH_3_	2.00 s	C:16, OAc	

*^a^* Recorded at 176.03 MHz in C_5_D_5_N/D_2_O (4/1). *^b^* Recorded at 700.00 MHz in C_5_D_5_N/D_2_O (4/1).

**Table 12 marinedrugs-18-00551-t012:** The cytotoxic activities of glycosides **1**–**7** and cladoloside C (positive control) against mouse erythrocytes, neuroblastoma Neuro 2a cells and normal epithelial JB-6 cells.

Glycoside	ED_50_, µM	Cytotoxicity EC_50_, µM	
	Erythrocytes	JB-6	Neuro-2a
Kuriloside A_1_ (**1**)	1.38 ± 0.07	2.81 ± 0.05	63.08 ± 2.42
Kuriloside A_2_ (**2**)	4.62 ± 0.15	5.87 ± 0.05	>100.00
Kuriloside C_1_ (**3**)	8.14 ± 0.01	45.24 ± 0.38	>100.00
Kuriloside D (**4**)	9.04 ± 0.54	11.58 ± 0.06	>100.00
Kuriloside E (**5**)	44.27 ± 0.80	96.80 ± 1.79	>100.00
Kuriloside F (**6**)	31.93 ± 0.44	49.81 ± 1.23	>100.00
Kuriloside A (**7**)	19.57 ± 0.32	23.02 ± 0.01	>100.00
Cladoloside C	0.31 ± 0.04	8.51 ± 0.12	11.92 ± 0.45

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
