# Peer review of "Kurilosides A1, A2, C1, D, E and F—Triterpene Glycosides from the Far Eastern Sea Cucumber Thyonidium (= Duasmodactyla) kurilensis (Levin): Structures with Unusual Non-Holostane Aglycones and Cytotoxicities"

_marinedrugs, 2020, doi:10.3390/md18110551_

Round 1
Reviewer 1 Report
Overall this was an interesting article, however, the manuscript requires major revision/editing due to its extensive grammatical errors. In some instances throughout the manuscript, "a" or "the" is missing. Also, the tense of the paper changes.
Please ensure that you define your abbreviations in full and then abbreviate the word. For example EtOH, should be written in full in the first instance and then abbreviated. Same for NMR, TOCSY, COSY, HSQC, ROESY, HR-ESI.
In the results and discussion section, actual methods have been extensively repeated. Results should contain only the results, not a repeat of methodologies used i.e. line 72-78. Please ensure that all tables are clearly referenced in your text, for e.g. please insert (Table 12) at the end of the sentence ending "were studied." (line 367). Also the sentence starting on line 366 and ending on 371 is far too long. There is too much information in the one sentence. Please amend by splitting the sentence into two.
As a general comment, hence and thus should not be used at the start of a new paragraph as they clearly indicate a continuance of discussion in the previous sentence/paragraph (see lines 182, 215, 257 and 351 and please amend).
With regards to MTT assay, the authors state that incubation was for 4-18 h (line 466). That is a large period of time. Exactly how long was the plate incubated for? Please be specific regarding incubation time. On line 484, please remove the comma after the word "that".
Author Response
Referee 1 commentaries
1) Overall this was an interesting article, however, the manuscript requires major revision/editing due to its extensive grammatical errors. In some instances throughout the manuscript, "a" or "the" is missing. Also, the tense of the paper changes.
2) Please ensure that you define your abbreviations in full and then abbreviate the word. For example EtOH, should be written in full in the first instance and then abbreviated. Same for NMR, TOCSY, COSY, HSQC, ROESY, HR-ESI.
3) In the results and discussion section, actual methods have been extensively repeated. Results should contain only the results, not a repeat of methodologies used i.e. line 72-78. Please ensure that all tables are clearly referenced in your text, for e.g. please insert (Table 12) at the end of the sentence ending "were studied." (line 367). Also the sentence starting on line 366 and ending on 371 is far too long. There is too much information in the one sentence. Please amend by splitting the sentence into two.
4) As a general comment, hence and thus should not be used at the start of a new paragraph as they clearly indicate a continuance of discussion in the previous sentence/paragraph (see lines 182, 215, 257 and 351 and please amend).
5) With regards to MTT assay, the authors state that incubation was for 4-18 h (line 466). That is a large period of time. Exactly how long was the plate incubated for? Please be specific regarding incubation time. On line 484, please remove the comma after the word "that".
The replies:
1) The manuscript was carefully checked and all necessary articles were inserted. The tense was equalized within the manuscript.
2) We absolutely disagree concerning abbreviations of the names of spectral methods such as NMR, TOCSY, COSY, HSQC, ROESY, HR-ESI. These abbreviations are too common and there is no necessity to provide their full names in each article of the journal. The abbreviation EtOH is also very trivial for chemical formulae. All these abbreviations are absolutely understandable for specialists and not interesting for all other. We should be very appreciative to the editors for the permission don’t describe the abbreviations listed above.
3) We have shortened the detail description of isolation procedures in the Results and Discussion. We have checked the references on the tables in the text and added the reference on the Table 12. The long sentence the referee noted was disrupted.
4) We have merged the paragraphs started with “hence” and “thus” with previous ones and replaced these words with any appropriated ones where it was impossible.
5) The time of incubation was replaced with certain value – 18 hr. The comma after “that” noted by the referee has been eliminated.
All the corrections are marked with yellow. We are very appreciative to the referee for careful checking of the manuscript.

Reviewer 2 Report
This work describes the characterization and biological activities of seven kurilosides from the Far Eastern sea cucumber Thyonidium (=Duasmodactyla) kurilensis (Levin).
It is an interesting study, which is well done and well written.
In my opinion, the paper can be accepted after minor modifications:
- The abstract should be shorted. The abstract should be a total of about 200 words maximum, and it is about 440 words.
- References should include DOI numbers.
- Typographical errors:
- Page 2, line 67: “data” instead of “dsta”
- Page 21, line 603: “Two” instead of “two”.
In summary, this paper includes so useful information but requires minor modifications to make it suitable for publication.
Author Response
The commentaries:
This work describes the characterization and biological activities of seven kurilosides from the Far Eastern sea cucumber Thyonidium (=Duasmodactyla) kurilensis (Levin).
It is an interesting study, which is well done and well written.
In my opinion, the paper can be accepted after minor modifications:
1) The abstract should be shorted. The abstract should be a total of about 200 words maximum, and it is about 440 words.
2) References should include DOI numbers.
3) Typographical errors:
- Page 2, line 67: “data” instead of “dsta”
- Page 21, line 603: “Two” instead of “two”.
In summary, this paper includes so useful information but requires minor modifications to make it suitable for publication.
The reply:
- The abstract is compressed.
- All the references are supplied by the doi where possible.
- The typographical errors are fixed.
All the corrections are marked with yellow. The authors are very appreciative to the referee for so careful checking of the manuscript.
